# Thermal pure matrix product state in two dimensions: tracking thermal equilibrium from paramagnet down to the Kitaev honeycomb spin liquid state

Matthias Gohlke[1], Atsushi Iwaki[2] and Chisa Hotta[2]

**1** Theory of Quantum Matter Unit, Okinawa Institute of Science and Technology Graduate University, Onna-son, Okinawa 904-0495, Japan

**2** Department of Basic Science, The University of Tokyo, Meguro-ku, Tokyo 153-8902, Japan and Komaba Institute for Science, The University of Tokyo, Meguro-ku, Tokyo 153-8902, Japan

August 3, 2023

## Abstract

We show that the matrix product state (MPS) provides a thermal quantum pure state (TPQ) representation in equilibrium in two spatial dimensions over the whole temperature range. We use the Kitaev honeycomb model as a prominent example hosting a quantum spin liquid (QSL) ground state to target the two specific-heat peaks previously solved nearly exactly using the free Majorana fermionic description. Starting from the high-temperature random state, our TPQ-MPS wrapping the cylinder precisely reproduces these peaks, showing that the quantum many-body description based on spins can still capture the emergent itinerant Majorana fermions in a $\mathbb{Z}_2$ gauge field. The truncation process efficiently discards the high-energy states, eventually reaching the long-range entangled topological state.

# 1 Introduction

Characterizing a thermal quantum state, a quantum many-body state at finite temperature is an ongoing fundamental challenge in condensed matter physics and beyond, since it is often a matter of quantum and classical correlations studied in statistical and quantum information physics [1]. Such a state has an intriguing aspect in that its representation is largely left facultative [2]; the Gibbs state is a mixture of an exponential number of states given by the density matrix $\rho_\beta$ of vanishing small purity $\mathcal{P} \sim e^{-\Theta(N)}$. The thermal pure quantum (TPQ) state, on the other hand, is a single pure state of purity $\mathcal{P} = 1$. In addition, there exist numerous thermal mixed quantum (TMQ) states with a purity between Gibbs and TPQ (see Fig. 1(a)). Canonical typicality guarantees that all these choices equivalently yield the same thermal equilibrium properties of the subsystem [3,4], and are macroscopically in the "same" thermal state. Since Gibbs, TPQ, and TMQ states rely on different design concepts, even when applying the "same" tensor network representation, its structure, convergence, or the amount of numerical resources required likely depend on which type of thermal state is chosen.

An important development concerning the Gibbs state is the matrix product density operator (MPDO), which provides a direct tensor network representation of the density matrix operator, $\rho_\beta$ [5,6]. Another standard form of the Gibbs state is the purified state analog to thermofield double, consisting of the size-$N$ system and the same numbers of ancilla degrees of freedom each suspended to a local site [7]. Ancilla serve as an entanglement bath and tracing out the ancilla corresponds to taking the Gibbs ensemble. These emphdoubled states also conform to a matrix product operator (MPO) approach [1], whose schematic illustrations are shown in Fig. 1(a). Here, the entanglement entropy is meaningless as a measure to characterize the Gibbs state. Instead, the thermal area law of mutual information between subsystems determines the bond dimension $\chi$ of MPO's [10–12]. The numerical drawback of MPDO or purification is the increase of the Hilbert space dimension due to the doubled degrees of freedom. Still, MPDO has been developed further recently using the XTRG algorithm [13], which realizes an exponential cooling down of the system by iteratively multiplying the matrix $\rho_\beta \times \rho_\beta = \rho_{2\beta}$, allowing to reach very low temperatures rapidly. XTRG has successfully been applied to two dimensions including our target [14, 15], the Kitaev honeycomb model [16].

The TPQ state, in comparison, consisting only of physical degrees of freedom, is pure by construction, and does not need the doubling of the local Hilbert space. In MPDO and its analogues, the doubling or the ancilla play the role of an ensemble average—or the classical mixture of states—which provide the volume-law thermal entropy. The lack of doubling implies that the pure TPQ state needs to store the same amount of entropy internally as a volume-law entanglement entropy [17–19]. For such purpose, the tensor-network-based representation bounded by the area law entanglement are thought to naturally be out of reach. Yet, the authors have recently exploited the specific form of matrix product state (MPS) practically recovering the volume law entanglement; only two ancilla/auxiliaries are attached to both edges of the one-dimensional (1D) MPS train, yet they have turned out to be sufficient to keep the nearly uniform distribution of entanglement entropy density throughout the system [2] which is essential for the volume law entanglement. We call this construction the TPQ-MPS [19].

---

[1]The difference between MPDO and purification is that the MPDO is not necessarily positive definite after truncation, whereas purification using a canonical form is positive definite. However, purification generally requires larger $\chi$ than MPDO [8], and there are some examples [9] that the purification MPO shows a divergence of $\chi$ at low temperatures, which may indicate that the thermal area law may not safely apply.

[2]If we take a bipartition of the TPQ-MPS system into left and right, each attached to the auxiliary, the entanglement entropy does not depend on the size of the left/right part, unlike the usual MPS that follows the size-dependent Page curve. This translational invariance of the entanglement entropy allows entanglement entropy between the center-$n$ sites and the rest (with $N - n$ sites and two auxiliaries) to follow the $n$-linear volume law (see Ref. [19]).

The TPQ state itself has a numerically long history [20–23] far before the formulative seminal works [24, 25]. They mostly rely on a full Hilbert space representation using Lanczos-based methods that limit the system size to typically $N \lesssim 30 - 40$. The TPQ-MPS largely shrinks the representation space and increases $N$ by factors by efficiently choosing its constituent states to those representing the target temperature limited by the bond dimension of the MPS.

The present work advances a few steps in developing a TPQ-MPS for two dimensions (2D), particularly for a quantum mechanically nontrivial quantum spin liquid state with long-range entanglement. Encoding the substantial amount of entanglement expected for QSL within an MPS or a tensor-network is generically a challenging task, although reported in the case of ground state [26–28]. Our result is the first to track the state by an MPS in the nearly pure form from the high-temperature random state down to the QSL with substantial entanglement between limited selection of basis states.

We finally refer to some TMQ-state-based approaches; the minimally entangled typical thermal state (METTS) [29, 30] mixes (takes an equal weight average of) a series of MPS generated from the Markov process. The quantum Monte Carlo designs a local product state basis to suppress the sign problem [31, 32], which are recently highlighted in combination with the iPEPS.

## 2 Construction of the TPQ-MPS state

We consider the standard imaginary-time evolution in generating the TPQ state at inverse temperature $\beta = 1/T$ given as

$$|\Psi_\beta\rangle = e^{-\frac{\beta}{2}\mathcal{H}}|\Psi_0\rangle = \mathcal{U}(\beta/2)|\Psi_0\rangle, \tag{1}$$

where $\mathcal{H}$ is the Hamiltonian of the system of interest, and the initial state $|\Psi_0\rangle$ representing an 'infinite-$T$' state is chosen as random, satisfying $\overline{|\Psi_0\rangle\langle\Psi_0|} \propto I$.

We now specify the construction of TPQ-MPS utilized here. The 1D tensor train of size-$N$ and bond dimension $\chi$ is prepared with auxiliary degrees of freedom added to both ends to provide an entanglement bath (Fig. 1(b)). Here, instead of the $\chi \times \chi$ form proposed in Ref. [19], each auxiliary consists of $N_{\text{aux}}$ sites with the same local Hilbert space $d$ as the physical sites of the system, i.e. $d = 2$ for $S = \frac{1}{2}$ spins, resulting in rank-3 tensors of the form $\chi_{i-1} \times \chi_i \times d$. The number of auxiliary sites dictates the maximum bond dimension at the edge of the physical system as $\chi_{\text{aux}} = d^{N_{\text{aux}}}$ and, hence, the maximum amount of entanglement between the auxiliary and the system. We emphasize that the auxiliary sites are not coupled to the physical system by any physical exchange, and therefore only the identity is applied to them during the imaginary time-evolution.

We extend TPQ-MPS to two spatial dimensions by wrapping the lattice on a cylinder with a finite circumference and wind the 1D MPS structure around, enumerating all the sites linearly (see Fig. 1(c). Cylinder tensor networks are fairly standard techniques nowadays, involving various variants in the way of wrapping the lattice and subsequent enumeration schemes. The precise way of wrapping the lattice can have physical implications; The system, although gapless in the two-dimensional limit, maybe gapped if the gapless nodes are not on allowed momenta lines in the Brillouin zone [27]. There are choices of particular cylindrical geometry known to capture the gapless state of the KH model [28, 33], but are not used here. The choice of such cylinder is important for the ground state but not for the temperature we can reach in the present study. The enumeration scheme should, ideally, not alter the physical properties. However, in reality, it can influence the spatial distribution of correlations and entanglement in particular for relatively small bond dimensions [14]. We employ a helical enumeration scheme with $8\times3\times2$ (YC8×3×2, which has circumference $L_{\text{circ}} = 6$ and is illustrated in Fig. 1(c)) and

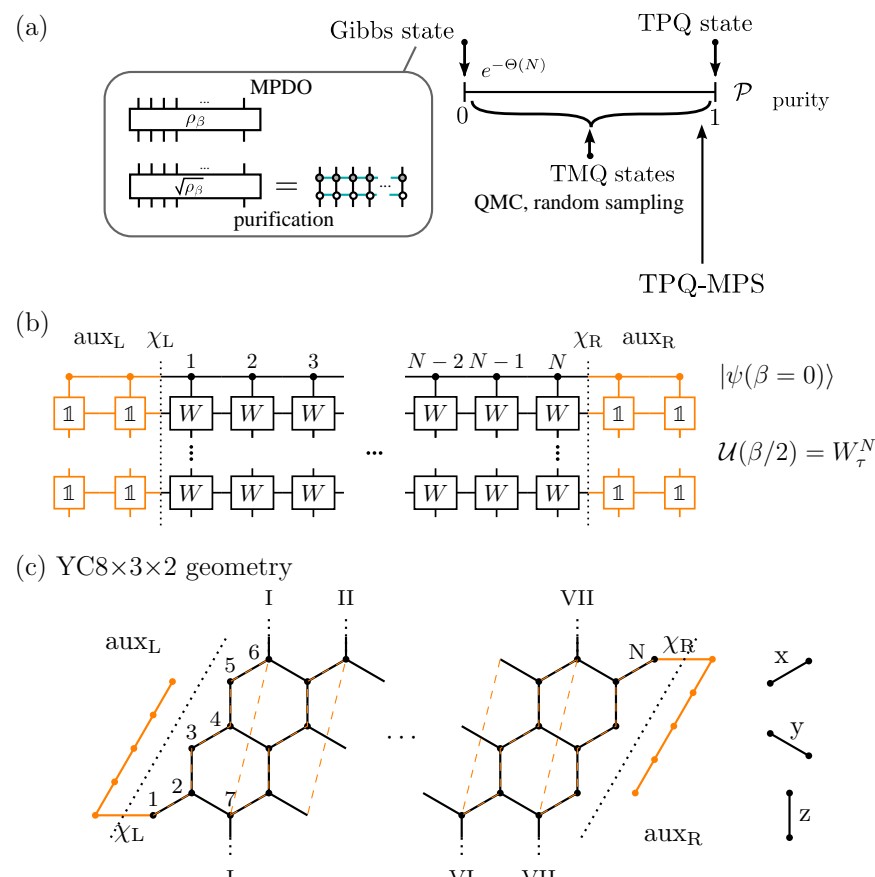

Figure 1: (a) Schematic illustration of different thermal quantum states, classified by purity. The two well-known representations of the Gibbs state are sketched. (b) Schematic illustration of TPQ-MPS with auxiliary degrees of freedom providing an entanglement bath, and the MPO-based imaginary time evolution. (c) Illustration of the honeycomb lattice composed of $x$-, $y$-, and $z$- bonds. For the underlying 1D MPS structure, we use a cylindrical geometry with a helical enumeration scheme as is highlighted by orange dashed lines. Equivalent bonds across the boundary are marked with the same roman literal. Specifically shown is the YC8×3×2 geometry with a shifted (by one lattice vector) boundary condition. Orange dots connected by solid lines represent the auxiliary sites.

109  $8 \times 4 \times 2$ sites (YC8×3×2, $L_{\text{circ}} = 8$) conforming to YC3-1 and YC4-1, respectively, using the
110  convention in Ref. [34]. Both schemes treat the $x$- and $z$-bond on equal footing, i.e. they are
111  nearest neighbors in the 1D MPS structure, while the $y$-bonds turn into an exchange with range
112  $2L_{\text{circ}} - 1$ sites. This choice results in the smallest $\chi_{MPO}$ of the time-evolution unitary, while
113  also reducing the number of nearest-neighbor bonds cut by a bipartition which, at sufficiently
114  low $T$, enters the amount of entanglement entropy encoded in the TPQ-MPS.

115     The long-range interactions within the effective 1D model make the time-evolving block
116  decimation scheme [35–37] in Eq. (1) infeasible. Instead, we rely on an MPO formulation of
117  the time-evolution operator [38] [3]. Specifically, we discretise $\mathcal{U}(\beta/2) = [\mathcal{U}(d\tau)]^N$ with small
118  imaginary time steps $d\tau$ and represent $\mathcal{U}(d\tau)$ as MPO [38]. [4] After each MPO-MPS product,

---

[3]We note that time-dependent variational principle (TDVP) [39,40] can be utilized as well.

[4]The MPO representation of the imaginary time evolution is given as $W^{\text{II}}(d\tau) \equiv \mathcal{U}(d\tau)$, following Ref. [38]. Splitting $d\tau = \tau_1 + \tau_2$ with sufficiently chosen complex $\tau_1$ and $\tau_2$ such that $\mathcal{U}(d\tau) \approx W^{\text{II}}(\tau_2)W^{\text{II}}(\tau_1)$ reduces the error in $d\tau$ by one order.

119  the MPS is compressed using a variational scheme [41] reducing $\chi$. We use an upper bound
120  for the maximum $\chi_{\text{bulk}} = \{362, 512, 724\}$ to limit the computational resources needed. If the
121  bound is not reached, small Schmidt values $\lambda_i$ are discarded provided either of the two criteria
122  are met: (I) discard all $\lambda_i \leq s_{\text{trunc}}$ or (II) discard all $\lambda_i$ sufficing $\sum_i \lambda_i^2 < (s_{\text{trunc}}^{\text{sum}})^2$ beginning
123  from the smallest $\lambda_i$.

124       Further technical details are given as follows; The initial random TPQ-MPS state $|\Psi_0\rangle$ is
125  prepared by applying a sequence of random unitaries to Néel-like product state in the $z$-basis,
126  i.e. $|\cdots \uparrow \downarrow \cdots\rangle$ [5] and cap the bond dimension at $\chi_{\text{ini}} = 32$. We prepare $N_{\text{samples}} = 100$ indepen-
127  dent random initial states (see Supplementary A for details [2]) and take the imaginary-time
128  step typically as $d\tau = 0.1$ and finer ones at $\beta \leq 0.8$. The truncation thresholds are set to
129  $s_{\text{trunc}} = 10^{-6}$ and $s_{\text{trunc}}^{\text{sum}} = 10^{-5}$ unless stated otherwise. Measurements are not independent
130  concerning $\beta$, but are done at certain series of fixed $\beta$ during the single run of imaginary-
131  time evolution and the $N_{\text{samples}}$ averages are taken from a set of independent runs. The TenPy
132  library [42] is used for all MPS-related numerical calculations.

## 3  Application to the Kitaev honeycomb model

134  We employ TPQ-MPS to the Kitaev honeycomb (KH) model defined as [16]

$$\mathcal{H} = K_x \sum_{\langle i,j\rangle_x} \sigma_i^x \sigma_j^x + K_y \sum_{\langle i,j\rangle_y} \sigma_i^y \sigma_j^y + K_z \sum_{\langle i,j\rangle_z} \sigma_i^z \sigma_j^z \,, \tag{2}$$

135  where $\sigma_i^\gamma$ are Pauli operators $\sigma^x$, $\sigma^y$, and $\sigma^z$ acting on sites $i$. The three sets of parallel bonds
136  on the honeycomb lattice are labeled as $\gamma = \{x, y, z\}$ (see Fig. 1(c)). The Kitaev interaction $K_\gamma$
137  couples a neighboring pair of spins $\langle i,j\rangle_\gamma$ along the $\gamma$-bond by an Ising-like exchange $\sigma_i^\gamma \sigma_j^\gamma$.
138  The KH model features a gapless QSL ground state if $K_\alpha \leq K_\beta + K_\gamma$ is satisfied for all per-
139  mutations of the bond labels $\{x, y, z\}$. Otherwise, a gapped QSL is found which adiabatically
140  connects to the Toric Code [43]. Here, we focus on the case of $K_x = K_y = K_z = \frac{1}{3}$.

141       The KH model features a double-peak structure in the specific heat, signalling crossovers
142  and releasing an entropy of $\Delta S/N = \frac{1}{2}\ln 2$ each. The associated two energy scales are well
143  known [44]: At the high-$T$ peak, $T_H/K \approx 0.5$, nearest-neighbor spin-spin correlations develop
144  and the fractionalization into itinerant and localized Majorana fermions occurs. The latter
145  contributes to the formation of fluxes at each hexagonal plaquette given as $W_{\mathcal{P}} = \prod_{i \in \mathcal{P}} \sigma_i^{\gamma_{\mathcal{P}}(i)}$,
146  where $\gamma_{\mathcal{P}}(i) = x, y, z$ is the label of bond connected to site $i$ while not being part of the
147  plaquette $\mathcal{P}$. The fluxes give an extensive set of quantum numbers, $w_{\mathcal{P}} = \pm 1$, which are
148  disordered at $T \lesssim T_H/K$. Below the low-$T$ peak, $T_L/K \approx 0.016$, the fluctuation of fluxes is
149  suppressed and we eventually find $\langle W_{\mathcal{P}}\rangle \to 1$. They form the static $\mathbb{Z}_2$ lattice-gauge field, fixing
150  half of the Hilbert space per unit cell. A local Hilbert space dimension of $\sqrt{2}$ per site remains
151  which is associated with itinerant-free Majorana fermions. Although the KH model at finite
152  temperature is not exactly solvable, once $\mathbb{Z}_2$ bond variables constituting the $\mathbb{Z}_2$ gauge field are
153  treated as classical degrees of freedom, a combination of classical Monte Carlo method with
154  free (Majorana) fermion exact diagonalization (MC+FFED) provides a nearly exact calculation
155  in a relatively large cluster, as performed by Nasu, *et.al* [44]. Whereas, its counterpart Eq.(2) is
156  a quantum many-body Hamiltonian which is generically difficult to solve at finite temperatures
157  from the front door. Therefore, the model provides a good platform and benchmark for our
158  approach. We would like to emphasize that our approach, unlike MC+FFED, is not custom
159  tailored to the Kitav model and can be applied to other quantum many-body Hamiltonian.

---

[5]We take 25 iterations of applying a random unitaries. The randomization is based on a TEBD algorithm with random two-site unitaries. Although not relevant to our model, an advantage of this method is the possibility of utilizing charge conservation, e.g. when studying a $U(1)$ symmetric model.

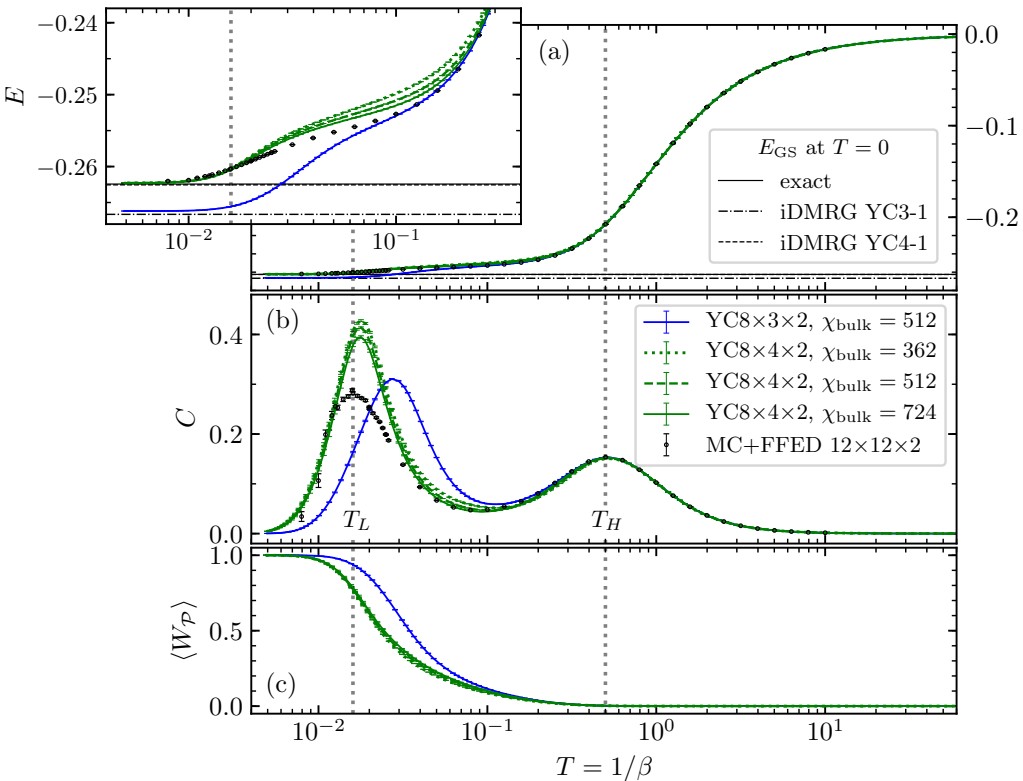

Figure 2: Temperature ($T = \beta^{-1}$)-dependent (a) energy density $E$ and (b) specific heat density exhibiting the double peak obtained by TPQ-MPS on cluster YC8×3×2 and YC8×4×2 and several $\chi_{\text{bulk}}$. Reference date (black dots) uses MC+FFED on $12 \times 12 \times 2$ sites [44]. YC8×4×2 exhibits a good quantitative agreement with MC+FFED down to $T \sim 0.05$. We attribute the difference in the position of $T_L$, in particular for YC8×3×2, to the finite circumference geometry used here; the ground state energies $E_{GS}$ of an infinitely long $L_{\text{circ}} = 6$ cylinder (YC3-1, dash-dotted horizontal line) obtained by iDMRG has a ground state energy density lower than the bulk exact one [16] (solid horizontal line) by about $\sim 0.01K$, yielding different crossover slopes in $E$ near $T_L$ and a shifted peak. (c) The evolution of the plaquette flux average, $\langle W_p \rangle$. The flux-free state at $T = 0$ exhibits $\langle W_p \rangle = 1$.

160    Our TPQ-MPS data in Fig. 2 exhibits a good qualitative agreement with the results obtained
161  from MC+FFED [44] on a $12 \times 12 \times 2$ cluster and XTRG using a YC6 × 4 × 2 geometry [15];
162  The energy density[6] $E$ rapidly decreases near $T_H$ resulting in a crossover peak in the specific
163  heat $C$. A second step of energy reduction occurs near $T_L$. The two-step behavior is already
164  present for small $\chi_{\text{bulk}} = 362$ with well converged behaviour down to $T \sim 0.2$ including the
165  high-$T$ peak in the specific heat. Whereas for $T \lesssim 0.2$, the finite-size and finite-$\chi$ effects
166  inevitably influence the data; In Fig. 2(a) we display in two different lines the ground state
167  energy obtained using iDMRG on an infinite cylinder with the same circumference $L_{\text{circ}} = 6, 8$
168  and helical boundary condition YC3-1 and YC4-1, respectively. The circumference seriously
169  affect the numerically achieved ground state energies and consequently the specific heat which
170  can be summarized as follows: (I) The cylinder with $L_{\text{circ}} = 6$ features an enhanced reduction
171  in energy upon cooling down approaching the significantly lower ground state energy. The
172  low-T peak in specific heat is of similar height to MC+FFED, while shifted to a two to three

---

[6]We are computing the energy density neglecting the left $N_l = L_{\text{circ}}$ and right $N_r = L_{\text{circ}}$ sites of the physical system to obtain a better estimate of the energy density in the bulk

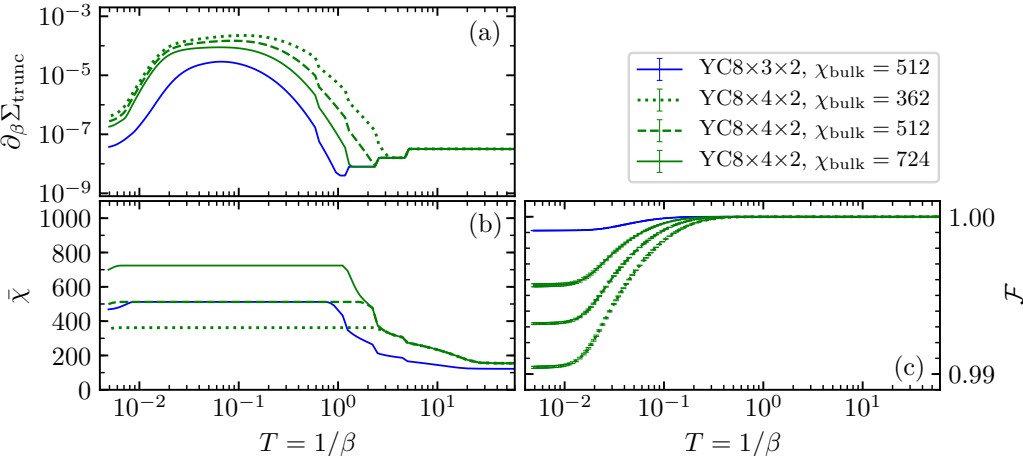

Figure 3: Evolution of the truncation error and bond dimension of the KH model: (a) Accumulated truncation error per unit of imaginary-time $\partial_\beta \Sigma_{\mathrm{trunc}}$ and bond dimension $\bar{\chi}$ averaged over the system, and (b) fidelity $\mathcal{F}$ of the evolved MPS before and after the truncation accumulated for all $\beta_j < \beta$.

times higher temperature. (II) For $L_{\mathrm{circ}} = 8$ we obtain an evolution of the energy closer to MC+FFED, thus reducing the finite-size effect signicantly. For $T_L \gtrsim T \gtrsim 0.2$, however, TPQ-MPS overestimates $E$ compared to MC+FFED. Here, increasing $\chi$ gradually reduces $E$ possibly approching MC+FFED for sufficiently large $\chi$. Near $T \sim T_L$ and below, the effect of finite $\chi$ ceases and the energy eventually approaches both MC+FFED as well as the ground state energy. As a consequence of the overestimated energy density at intermediate $T$, we obtain an enhanced slope of $E$ resulting in a higher peak in the specific heat. Again, increasing $\chi$ improves accuracy, reduces the height of the peak and results in a behaviour closer to MC+FFED. The peak position is very similar to MC+FFED at any $\chi$.

A recent XTRG calculation applied to the Kitaev model reports the lower-peak at $T_L \sim 0.19$ with the peak-height of $\sim 0.3$ using a $6 \times 4 \times 2$ cylinder [15]. While the circumference is similar to our YC8$\times$4$\times$2, the XTRG work uses a slightly shorter cylinder, does not use helical boundary condition, and employs a different winding scheme. The agreement with XTRG is very good above $T \sim T_L$, but while $T_L$ is similar to XTRG, we obtain higher peaks.

The average of $\mathbb{Z}_2$ fluxes in Fig. 2(c) nicely marks the two peaks by an onset of nonzero value ($T_H$) and the inflection point ($T_L$), finally approaching $\langle W_{\mathcal{P}} \rangle \to 1$ at $T \to 0$ systematically for various $\chi_{\mathrm{bulk}}$.

We like to remark that in many frustrated spin models, the specific heat at $T \lesssim 0.1$ naturally suffers large finite-size effect independent of the method employed. For example, in kagome lattice Heisenberg antiferromagnet, specific choices of clusters sometimes yield unphysical peaks or features not observed in other choices of cluster [25, 45] possibly obscuring the physical behaviour.

# 4 How truncation affects the TPQ-MPS state

We now quantify the TPQ-MPS based on the error analysis during the run by focusing on two quantities: The first one is the sum of all discarded Schmidt values ($\lambda_i$ for $i > i_0(\beta_j)$ which fulfills aforementioned I or II in §.2) accumulated over a single imaginary-time evolution,

$$\Sigma_{\mathrm{trunc}}(\beta) = \sum_{\beta_j < \beta} \sum_{i > i_0(\beta_j)} \lambda_i^2(\beta_j) . \tag{3}$$

The second one is the product of the fidelities of the state $|\Psi(\beta_j)\rangle = W^{\mathrm{II}}(d\tau)\Psi(\beta_{j-1})\rangle$ (see Ref.[36]) and $|\tilde{\Psi}(\beta_j)\rangle$ just before and after truncation, respectively, for all truncations down to the temperature $\beta^{-1}$,

$$\mathcal{F}(\beta) = \prod_{\beta_j < \beta} (|\langle \tilde{\Psi}(\beta_j)|\Psi(\beta_j)\rangle|^2), \tag{4}$$

which evaluates how we deviate from the non-truncated wave function at $\beta$. The amount of truncated Schmidt values per unit of imaginary time is given as $\partial_\beta \Sigma_{\mathrm{trunc}}$. In Figure 3 we show the evolution of $\partial_\beta \Sigma_{\mathrm{trunc}}$, of the average bond dimension $\bar{\chi}$, and of $\mathcal{F}$. Upon cooling down, $\partial_\beta \Sigma_{\mathrm{trunc}}$ remains below $10^{-7}$ until $\bar{\chi}$ reaches $\bar{\chi} \sim \chi_{\mathrm{bulk}}$, which occurs near $T \sim 1$. Larger $\chi_{\mathrm{bulk}}$ (smaller system) generally lowers this threshold temperature. At these high temperatures, the evolution is very accurate reflected in a fidelity $\mathcal{F} \sim 1$. Upon lowering the temperature, $\partial_\beta \Sigma_{\mathrm{trunc}}$ increases and then reaches a plateau at $T_L \lesssim T \lesssim T_H$ with values $\partial_\beta \Sigma_{\mathrm{trunc}} \sim 10^{-5}$ to $10^{-4}$ depending on $\chi_{\mathrm{bulk}}$. Here, $\mathcal{F}$ starts to depart gradually from 1, which is more distinct for smaller $\chi_{\mathrm{bulk}}$. At $T \lesssim T_L$ the error $\partial_\beta \Sigma_{\mathrm{trunc}}$ reduces again and $\mathcal{F}$ starts to flatten out. For YC8×3×2 at $\chi_{\mathrm{bulk}} = 512$, $\bar{\chi}$ slightly drops, indicating the reduction in the size of the Hilbert space needed to effectively encode the low-temperature state.

These observations suggest two effects of the truncation $\chi$; For relatively small $\chi_{\mathrm{bulk}}$ that is reached quickly, in particular at intermediate $T_L \lesssim T \lesssim T_H$, taking a larger $\chi$ lowers the energy towards the optimal value. This becomes evident upon inspection of $E$ in Fig. 2(a), whose accuracy improves for larger $\chi_{\mathrm{bulk}}$ approaching the MC+FFED data.

The second effect concerns the states at high energy. Let us expand the TPQ state constructed for the full Hilbert space for finite $N$. The system is split into a smaller part $A$ (with dimension $D_A$) and a bigger part $B$, which is Schmidt decomposed as

$$|\Psi_\beta\rangle = \sum_{n=1}^{D_A} \lambda_n |n_A\rangle |n_B\rangle \tag{5}$$

to the orthogonal basis sets $\{|n_A\rangle\}$ and $\{|n_B\rangle\}$. The local part $A$ is thermalized and its density operator is approximated by the Gibbs state in $A$ as

$$\rho_A = \sum_{n=1}^{D_A} \lambda_n^2 |n_A\rangle\langle n_A| \simeq \frac{e^{-\beta \mathcal{H}_A}}{Z_A} \tag{6}$$

where $\{|n_A\rangle\}$ is thought to be the energy eigenbasis of the subsystem's Hamiltonian $\mathcal{H}_A$. For its eigenvalues $\{E_n^A\}$, the Schmidt coefficient $\lambda_n$ is represented as $e^{-\beta E_n^A/2}/\sqrt{Z_A}$, and we find

$$|\Psi_\beta\rangle \simeq \sum_{n=1}^{D_A} \frac{e^{-\beta E_n^A/2}}{\sqrt{Z_A}} |n_A\rangle |n_B\rangle. \tag{7}$$

Note here that $\{|n_B\rangle\}$ is left unknown. We finally truncate $|\Psi_\beta\rangle$ as $D_A \to \chi$ in Eq.(7), discarding the basis states with small weight. Specifically, information of $|n_A\rangle$ belonging to higher $E_n^A$ is lost. This explains the capability of TPQ-MPS to express qualitatively different quantum states from high to low temperatures; The truncation of the MPS efficiently compresses the information needed to represent the thermal state in particular at low temperatures. Moreover, in TPQ-MPS, the variance of physical quantities among different initial states becomes smaller by more than one order for lower temperature [2, 19]. This is in sharp contrast to the usual random sampling methods or Monte Carlo methods, where the sampling error is by orders of magnitude larger in the lower temperature phase.

## 5   Conclusion

To summarize, the TPQ-MPS is applied to 2D by wrapping the MPS train into cylinders. The two peaks in the specific heat in the Kitaev honeycomb lattice signaling the fractionalization of spins into Majorana fermions and fixing the $\mathbb{Z}_2$ gauge flux are both reproduced. While finite-size effects appear at $T \lesssim 0.1$ as is common with other methods, finite-$\chi$ affects the MPS-TPQ only intermediate temperatures $T \sim 0.1$ and is less of a concern at very low temperatures $T \lesssim 0.01$. Here, the truncation process of TPQ-MPS efficiently discards the higher-temperature information explaining why it can track a nearly pure thermal state with its volume-law entanglement–equivalent to the thermal entropy–across a wide range of temperatures. This allows the state starting from random at high temperature (initial state) to gradually reach the qualitatively different long-range entangled topological ordered ground state.

## Acknowledgements

We thank J. Nasu for providing us with reference data. We acknowledge the use of computational resources of the supercomputer Fugaku provided by the RIKEN AICS through the HPCI System Research Project (Project ID: hp210321) and of the Scientific Computing section of the Research Support Division at the Okinawa Institute of Science and Technology Graduate University (OIST). M.G. acknowledges support by the Theory of Quantum Matter Unit at OIST.

**Funding information**    This work was supported by a Grant-in-Aid for Transformative Research Areas "The Natural Laws of Extreme Universe— A New Paradigm for Spacetime and Matter from Quantum Information" (No. 21H05191) and JSPS KAKENHI (Grants No. JP21K03440 and JP22K14008). A.I. was supported by JSPS Research Fellowship (Grant No. 21J21992).

## A   Random sampling average

TPQ-MPS is a random sampling method using the MPS representation of the quantum many-body wave function. Since the quality of the MPS state relies its bond dimension $\chi_{\text{bulk}}$ practically accessible in the computation, a smaller $\chi_{\text{bulk}}$ requires a number of independent runs to be averaged over. This number is generally by orders of magnitude smaller than METTS and other MPS methods when applying them to the same system. One may anticipate that the present 2D Kitaev honeycomb (KH) model may require more samples than for 1D systems [19]. As illustrated in Fig. 1 in the main text, the higher the purity is, fewer samples $N_{\text{samp}}$ are needed to safely reproduce the thermal quantum state. In TPQ-MPS, the $N_{\text{sample}}$-independent runs are performed starting from the independent initial random MPS, yielding a set of *unnormalized* states over different $\beta_j$ for each, $\{|\Psi^{(l)}(\beta_j)\rangle\}_{l=1}^{N_{\text{sample}}}$. The random average of physical quantities $\mathcal{O}$ is taken as

$$\langle \mathcal{O} \rangle = \frac{\sum_{l=1}^{N_{\text{sample}}} \langle \Psi^{(l)}(\beta_j)|\mathcal{O}|\Psi^{(l)}(\beta_j)\rangle}{\sum_{l=1}^{N_{\text{sample}}} \langle \Psi^{(l)}(\beta_j)|\Psi^{(l)}(\beta_j)\rangle}. \tag{8}$$

Here, the summations of samples are taken independently between the numerator and denominator, since the partition function is given by the denominator $Z = \sum_{l=1}^{M} \langle \Psi^{(l)}(\beta_j)|\Psi^{(l)}(\beta_j)\rangle$ (the reasoning for why the average taken by the normalized $|\Psi^{(l)}(\beta_j)\rangle$ does not provide the correct sampling average is analytically shown in Ref.[ [2]]).

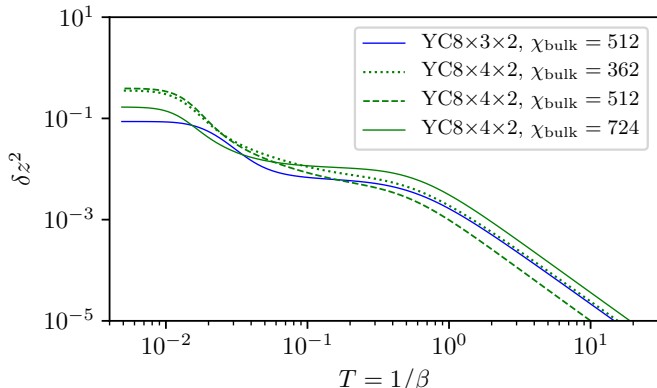

Figure 4: Normalized fluctuation of partition function (NFPF) $\delta z^2$ as a function of $T$ for the data calculated in Fig. 2 for the KH model with different $\chi_{\text{bulk}}$. The number of samples required to obtain the same quality data scales linearly with $\delta z^2$.

The number $N_{\text{sample}}$ required can be measured using a quantity called normalized fluctuation of partition function (NFPF),

$$\delta z^2 = \frac{\text{Var}(\langle \Psi(\beta_j)|\Psi(\beta_j)\rangle)}{\left(\overline{\langle \Psi(\beta_j)|\Psi(\beta_j)\rangle}\right)^2},\tag{9}$$

where $\overline{\cdots}$ is the random average. It is shown that the purity of the thermal state scales with $\delta z^2$ and the larger $\delta z^2$ means that the obtained state varies much with a sample. In fact, we showed that the number of samples needed, $N_{\text{sample}}$, to obtain the same quality of Eq.(8) increases proportionally to $\delta z^2$ [2].

The results of $\delta z^2$ for the present calculation on the KH model are given in Fig. 4 for a set of data given in Fig. 2. The largest $\delta z^2$ at low-$T$ ranges at $10^{-1}-10^1$, which is comparable to the value for the 1D Heisenberg model [2] which used $N_{\text{sample}} = 100$. Based on this comparison, we also adopt $N_{\text{sample}} = 100$ for the 2D case. The present calculation shows that the 2D TPQ-MPS is as capable as the 1D case despite the consensus that the calculations in 2D are much more difficult than in 1D.

The plateau of $\partial_\beta \Sigma_{\text{trunc}}$ observed in Fig. 3 agrees with the plateau of $\delta z^2$, and as in Fig. 3, $\chi_{\text{bulk}}$ dependence appears at $T \lesssim 10^0$, showing that $\delta z^2$ is indeed a good measure to qualify the quantum state. We find a suppression of $\delta z^2$ by a log scale in terms of $\chi_{\text{bulk}}$, indicating the high capability of TPQ-MPS to store the information required for a wide range of temperatures exhibiting different natures.

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
