# Peer review of "Thermal pure matrix product state in two dimensions: tracking thermal equilibrium from paramagnet down to the Kitaev honeycomb spin liquid state"

_SciPost Physics_

## Round 1 · Referee Report · Anonymous (Referee 2) · 2023-9-11

Strengths

1) State of the art numerical many body method to treat quantum and thermal fluctuations on equal footing reaching large system sizes.

2) Benchmark results on the spin-1/2 Kitaev Honeycomb model give accurate results for two-peaked structure of the specific heat, including the positions of the two peaks and the height of the second peak.

Weaknesses

1) A more clear discussion on the strengths and limitations of the TPQ-MPS method would be beneficial.

2) A discussion on the applicability of the TPW-MPS method to non-frustrated systems would be beneficial.

Report

The Authors study the spin-1/2 Kitaev model on the Honeycomb lattice using TPQ states in the MPS representation. The results reproduce the two-peak structure of the specific heat, in particular the positions of the two peaks and the height of the high-T peak. The disagreement in the low-T peak height is attributed to finite-size effects, which are also visible in the comparison of the calculated ground state energies with the exact results for the bulk system.

Overall these results confirm that TPQ states can reproduce thermodynamics in a wide T range, down to very low temperatures in a system with long-range entanglement.

These are in agreement with previous works using TPQ states (or their finite-T Lanczos extensions where an averaging over several TPQ states is taken in the end) in highly frustrated magnets, including kagome and kagome-like magnets, and the Kitaev honeycomb model, see, e.g., Phys. Rev. B 93, 174425 (2016), Phys. Rev. B 100, 045117 (2019), Phys. Rev. B 105, 144427 (2022), Phys. Rev. B 107, 245115 (2023).

The method is interesting and provides another step towards the development of accurate quantum many body methods that treat quantum and thermal fluctuations on equal footing.

Requested changes

The Authors could consider the following suggestions:

1) The Authors could provide a discussion on the applicability of this method to non-frustrated systems, where one expects finite-size effects to set in at higher temperatures due to the very discrete nature of the spectrum of finite-size systems (as opposed to the dense spectrum of highly frustrated systems down to very low energies).

2) The first sentence of the abstract (“… the matrix product state (MPS) provides a thermal quantum pure state  (TPQ) representation in equilibrium in two spatial dimensions over the whole temperature range.”) is unclear and/or confusing. Do the Authors intend to highlight that MPS states can now be used for making predictions at nonzero temperatures? If so, then the Authors could strengthen this aspect and discuss more explicitly the added advantages of using the MPS representation on reaching larger system sizes, as opposed e.g. to the full basis representation (standard ED), but also the possible disadvantages (artificial boundary effects due to quasi-1D geometries).

3) further minor comments:

line 29: Theta and N not defined

line 42: “emphdoubled” unclear

line 85: the bar and symbol “I” in “\bar{|\Psi_0\rangle \langle\Psi_0|} \propto I “ not defined.

line 157: “from the front door” unlcear

  • validity: high
  • significance: high
  • originality: high
  • clarity: good
  • formatting: excellent
  • grammar: good

Author:  Matthias Gohlke  on 2023-10-20  [id 4045]

(in reply to Report 2 on 2023-09-11)
Category:
answer to question

We like to thank the Referee carefully reading our manuscript and for evaluating our method as state-of-art. We have revised our manuscript according to the referees helpful comments. Please find our detailed reply below.

These are in agreement with previous works using TPQ states (or their finite-T Lanczos extensions where an averaging over several TPQ states is taken in the end) in highly frustrated magnets, including kagome and kagome-like magnets, and the Kitaev honeycomb model, see, e.g., Phys. Rev. B 93, 174425 (2016), Phys. Rev. B 100, 045117 (2019), Phys. Rev. B 105, 144427 (2022), Phys. Rev. B 107, 245115 (2023).

1) The Authors could provide a discussion on the applicability of this method to non-frustrated systems, where one expects finite-size effects to set in at higher temperatures due to the very discrete nature of the spectrum of finite-size systems (as opposed to the dense spectrum of highly frustrated systems down to very low energies).

First, to be brief, TPQ-MPS is different from full TPQ and should perform at least equally well for non-frustrated as for frustrated systems. The referee points to successful applications of full TPQ method to highly frustrated magnets. The issue, however, is that for a full TPQ state at limited size-$N$ the sample-dependent variance (error) is of order $\le e^{-S/T}$ and increases significantly at low temperatures. This variance is suppressed in frustrated magnets where the system retains a high entropy $S$ down to low temperature, as pointed out analytically in the original paper Ref. [25]. Therefore, the variance becomes large for non-frustrated systems. This fact does not depend on the spatial dimensionality, and supports the Referee's statement on the advantage of full TPQ for frustrated systems over non-frustrated system.

Some of the authors have shown in Ref. [19], that for TPQ-MPS the tendency is opposite to full TPQ: The low temperature accuracy improves significantly and TPQ-MPS exhibits almost no sample dependence. In Sec. 4 on "How truncation affects the TPQ-MPS state" we provide further insight. Intuitively, this is because MPS provides a good description for a low temperature quantum state by discarding high energy states. We add a comment in Sec. 4 linking this fact to what is known about the full TPQ in frustrated systems.

Here, we also like to draw the Referee's attention to two of our publication on the kagome lattice Heisenberg model: [Asano, Hotta, Phys. Rev. B 98,140405, 2018] using full TPQ method (but as further advantages: free of size effect), and [Endo, Hotta, Shimizu, Phys. Rev. Lett.121,220601, 2018], obtaining the almost exact dynamical responses using dynamical TPQ. We have added a comment on this issue in our draft as well as the corresponding references.

2) The first sentence of the abstract ("… the matrix product state (MPS) provides a thermal quantum pure state (TPQ) representation in equilibrium in two spatial dimensions over the whole temperature range") is unclear and/or confusing. Do the Authors intend to highlight that MPS states can now be used for making predictions at nonzero temperatures? If so, then the Authors could strengthen this aspect and discuss more explicitly the added advantages of using the MPS representation on reaching larger system sizes, as opposed e.g. to the full basis representation (standard ED), but also the possible disadvantages (artificial boundary effects due to quasi-1D geometries).

We understand that our way of putting the abstract links to the Referee's comment on the weakness, "A more clear discussion on the strengths and limitations of the TPQ-MPS method would be beneficial.". Indeed, we have applied TPQ-MPS only to the Kitaev model in 2D, and although this model is considered not to be the easiest model to make accurate results, it is true that our statement seemed too general and may require more demonstrations. We have weakened this point and clarify that our work is a first successful application to a system in two spatial dimensions.

Regarding its general applicability, and as we mention in the previous reply, we do expect TPQ-MPS to have no disadvantage on non-frustrated systems. One advantage of TPQ-MPS is that it can be applied to larger systems than full TPQ also in 2D. For the ground state, even though the cylinder construction may still give serious cluster-size/shape dependence on the result, the MPS-based methods have been applied to a wide range of models, because larger system sizes than with full ED are possible. Our work clarified that the finite-temperature MPS-based method in 2D can be as applicable as those for the ground state. We would like to stress this point, and have added a remark to the final part of the abstract.

3) further minor comments. line 29: Theta and N not defined, line 42: “emphdoubled” unclear, line 85: the bar and symbol “I” in "$\bar{|\Psi_0\rangle \langle\Psi_0|} \propto I$” not defined. line 157: “from the front door” unclear

We thank the referee for spotting typos and bringing up ambiguous phrases. What we intended to highlight with the phrase ``from the front door'' is that treating the Kitaev model in the spin representation without the prior knowledge about the static $\mathbb Z_2$ gauge field is as much of a difficult problem as solving a generic quantum many-body system.

---

## Round 1 · Referee Report · Anonymous (Referee 1) · 2023-9-11

Strengths

1- first application of a promising MPS-based finite temperature approach to a 2D system 2- Benchmark of an interesting and non trivial problem hosting a quantum spin liquid

Weaknesses

  • The approach in 2D is tested only for one particular model. It would have been interesting to test it also for another model where data from METTS is available for a direct comparison
  • The system size of the reference data is different, making a direct comparison more tricky

Report

In this paper the authors apply the recently developed TPQ-MPS approach in two dimensions, taking the Kitaev honeycomb model hosting a quantum spin liquid as an example. Their approach reproduces the two specific heat peaks which have been previously found based on a free Majorana fermionic description. They compare their results also with previous studies based on XTRG, and provide a detailed discussion of the involved errors in their approach.

The study of thermal properties of frustrated spin systems is a great challenge, particularly in two dimensions. This work offers interesting insights into the applicability of the TPQ-MPS approach for these systems, which forms a valuable basis for future studies of challenging 2D systems based on this approach, as well as for generalizations to e.g. higher-dimensional tensor networks. For these reasons I can recommend publication of this paper in SciPost Physics, after addressing the points listed below.

List of questions and comments

  • On page 6 the authors mention that the TPQ-MPS data in Fig.2 exhibits a good qualitative agreement with MC+FFED and with XTRG, but Fig 2 does not show any XTRG data. If possible, it would be good to add it. Also, the authors mention that they obtain higher peaks than found in XTRG. It would be interesting to discuss the reason for this discrepancy.

  • In the footnote on page 5 it is written that 25 iterations of applying random unitaries are taken. What is the motivation for this choice?

  • Are there any advantages of using N_{aux} spins of dimension d rather than one "large" ancilla site of dimension d^N_{aux}? I guess it is equivalent but just simpler to implement on the technical level?

  • page 9, Appendix A: the authors write that the number of independent runs to be averaged over is smaller in TPQ-MPS than in METTS, but how about the required bond dimension? Do the authors have any insights into this?

  • Fig 1a: Why is there a sqrt(\rho_\beta) in the lower picture representing the purification and not just \rho_\beta?

  • minor typo: in the formula in the top line of page 8 there is a "|" missing before \Psi

  • validity: good
  • significance: high
  • originality: high
  • clarity: high
  • formatting: good
  • grammar: good

Author:  Matthias Gohlke  on 2023-10-20  [id 4046]

(in reply to Report 1 on 2023-09-11)
Category:
remark
answer to question

We thank the Referee for carefully reading our manuscript and for recommending our work for publication in SciPost Physics provided we address the questions raised. For these questions, we give a detailed reply below.

1. On page 6 the authors mention that the TPQ-MPS data in Fig.2 exhibits a good qualitative agreement with MC$+$FFED and with XTRG, but Fig 2 does not show any XTRG data. If possible, it would be good to add it. Also, the authors mention that they obtain higher peaks than found in XTRG. It would be interesting to discuss the reason for this discrepancy.

We thank the referee for the suggestion. We tried to contact the authors of Ref. [14] (Shou-Shu Gong, Yang-Qi and Wei Li) to provide the data, but did not obtain a reply. Instead, we have extracted the data from Fig.4 in Ref.[14] [Phys. Rev. B 100, 045110 (2019)] at $h=0$ which will have some ambiguity which we estimate as $|dC_V|<0.005$ and $|dT|<0.006$ (the size of the symbol of the original data). Nonetheless, this data should still contain sufficient information to allow for a direct comparison with our data and MC$+$FFED. We have updated Fig.2(b) to include the XTRG data. Generally, all three methods agree well at temperatures above $T \sim 0.1$ and deviations become apparent at lower temperature. In the XTRG data, the peak height at $T_L$ is comparable to MC$+$FFED, but the peak position shifts to higher temperature. This shift can be attributed to the different geometry containing $6\times 4\times 2$ sites and a different boundary condition. Although the circumference is similar to our YC8$\times$4$\times$2 cylinder, the different periodic boundary condition will affect the ground state energy as well as the spectrum and will result in a shift of the low temperature peak. This shift in the ground state energy and how it affects $T_L$ is nicely illustrated by our two geometries.

2. In the footnote on page 5 it is written that 25 iterations of applying random unitaries are taken. What is the motivation for this choice?

Our motivation for this choice is to prepare an initial random MPS that is sufficiently random, and storing as much entanglement as possible (subject to $\chi$) to represent the high temperature state. We have chosen $N=25$ iterations of applying random unitary matrices based on the evolution of the entanglement entropy of a bipartition, $S_E$. In Figure~4(b) in the draft (newly added), we plot the average entanglement $\bar{S_E}$ of the initial MPS to see how it evolves as a function of iterations. While the bond dimension of the state generally doubles after each iterations, $\bar{S_E}$ is not fully saturated after $N=5$ iterations which corresponds to $\chi_\text{RMPS} = 32$. Instead, saturation is reached after around $N\sim 10$ iterations. Our choice of $N=25$ iterations is well within the saturated regime. We have moved the footnote regarding the initial state preparation to the Appendix A, added Fig. 4(b), and expanded the explanation.

3. Are there any advantages of using $N_{aux}$ spins of dimension $d$ rather than one "large" ancilla site of dimension $d^{N_{aux}}$? I guess it is equivalent but just simpler to implement on the technical level?

There is no advantage of using $N_\mathrm{aux}$ spins of dimension $d$ over using a single degree of freedom with $d^N_\mathrm{aux}$ other than the ease of implementation and the physical intuition about the degrees of freedom included in the bath. We have added footnote 3 in the main text to address this.

4. page 9, Appendix A: the authors write that the number of independent runs to be averaged over is smaller in TPQ-MPS than in METTS, but how about the required bond dimension? Do the authors have any insights into this?

Let us quickly summarize some aspects of METTS: METTS starts from a product state with bond dimension $\chi=1$. $\chi$ grows upon performing the imaginary time evolution. METTS then uses a Markov chain scheme to sample over many initial product states. This implies that METTS is the mixed state with relatively small entanglement---and, thus, small bond dimension---for each constituent pure state. TPQ-MPS, on the other hand, starts from an initial random MPS with finite bond dimension, storing a sufficient amount of entanglement internally. In particular at higher temperatures, we would expect for $T \gtrsim J$, METTS requires a large sample average to obtain sufficient thermal entropy. TPQ-MPS on the other hand stores this entropy internally in form of entanglement and reduces the number of samples required drastically to the point where, potentially, no sample averaging is required. Indeed, there a trade off between the number of samples required and the bond dimension. For more details, please find our paper, Ref.[2], that analytically shows these relationships in more detail. We have added a comment in the appendix A to provide some insight into how TPQ-MPS and METTS should compare.

*5. Fig 1a: Why is there a $\sqrt{\rho_\beta}$ in the lower picture representing the purification and not just $\rho_\beta$? *

This is because purification has system sites and auxiliary degrees of freedom, and is realized by operating $\sqrt{\rho_\beta} \otimes U_{\mathrm aux}$ to the initial state, which is a product state of singlets consisting of $i$-th system site its auxiliary site. Here, $\sqrt{\rho_\beta} =e^{-\beta H/2}$ is the imaginary time evolution and $U_{\mathrm aux}$ is the unitary operator operating on system and auxiliaries, respectively. We have added a brief explanation in the caption of Fig. 1.

6. minor typo: in the formula in the top line of page 8 there is a "$|$" missing before $\Psi$

We thank the referee for spotting this typo. It has been revised accordingly.

---

## Editorial Decision

resubmitted